# Modeling of the Magnetic Turbulence Level and Source Function of Particle Injection from Multiple SEP Events

Lele Lian [1], Gang Qin [1,*], Shuangshuang Wu [1], Yang Wang [1] and Shuwang Cui [2,*]

1   School of Science, Harbin Institute of Technology, Shenzhen 518055, China
2   College of Physics, Hebei Normal University , Shijiazhuang 050024, China
*   Correspondence: qingang@hit.edu.cn (G.Q.); cuisw@mail.hebtu.edu.cn (S.C.)

**Abstract:** Solar energetic particles (SEPs) are produced by solar eruptions and are harmful to spacecraft and astronauts. The four source function parameters of particle injection for SEP events and the magnetic turbulence level can be collectively referred to as key parameters. We reproduce the electron intensity-time profiles with simulations for five SEP events observed by multispacecraft such as ACE, STEREO-A, and STEREO-B, so we can obtain the five fitted key parameters for each of the events. We analyze the relationship among the five fitted key parameters, and also the relationship between these parameters and the observed event features. Thus, the model of key parameters are established. Next, we simulate another 12 SEP events with the key parameters model. Though the predicted electron intensity-time profiles do not fit the observed ones well, the peak flux and event-integrated fluence can be predicted accurately. Therefore, the model can be used to estimate the radiation hazards.

**Keywords:** solar flares; solar energetic particles; interplanetary turbulence; magnetic fields; forecasting models

## 1. Introduction

Solar energetic particles (SEPs), which are produced by solar eruptions, can impact the interplanetary and near-Earth environments and damage spacecraft, astronauts' health, and interfere with advanced techniques such as navigation and communication (e.g., [1–3]). Generally speaking, SEP events can be divided into two groups, according to the characteristics of their sources (e.g., [4–6]): impulsive events and gradual events. Impulsive SEP events, which has the characteristics of short duration, low intensity, and rich in electrons, $^3$He, and heavy ions, are associated with soft X-ray (SXR) flares. In contrast, gradual SEP events, with the characteristics of long duration, high intensity, and richness in protons, are related to the shocks driven by coronal mass ejections (CME).

Predictions of SEP events, based on the understanding of the acceleration and propagation mechanisms of SEPs, are essential for reducing the potential radiation damages of SEPs. In the upcoming solar maximum of solar cycle 25, which is expected to occur in 2024–2025, we would witness a lot of SEP events. Therefore, predictions of SEP events are more crucial in the next few years, and these SEP events would provide a chance to validate and improve the prediction models.

Researchers investigate the relationship between the properties of SEP events and characteristics of flares or CMEs. To study the relationship, some researchers focus on flares [7,8]. For example, Kurt et al. [9] found that the yearly number of proton events is correlated with that of SXR flares with importance >M4, and the correlation coefficient (CC) is equal to 0.81. For the ground-level enhancement event of SEPs, Wu and Qin [10] found that the fluence of SXR flare is correlated with a spectral parameter, which is related to the fluence of SEPs, of the energy spectrum, so that the fluence of SXR flare is an important parameter for estimating SEP fluence. On the other hand, some researchers



focus on CMEs [11–13]. Finally, there are also some researchers who focus on both flares and CMEs [14–18]. With statistical analysis, Dierckxsens et al. [15] found that the SEP occurrence probability and proton peak fluxes at 1 au depend on the characteristics of both flares (intensity and longitude) and CMEs (speed and angular width).

Multispacecraft observations can provide more information for investigating SEP properties than single spacecraft observations. Lario et al. [19] found that the peak intensity depends on the longitude and can be approximately described by the Gaussian function through multispacecraft observations of SEP events by at least two spacecraft located near 1 au. Similar results were found by research on multispacecraft observations (e.g., [20–22]). In addition, the dependence of SEP intensity-time profiles on the location of the solar source region relative to the observer were investigated according to multispacecraft observations (e.g., [19,23–25]). In addition, Richardson et al. [18] studied the properties of ∼25 MeV protons and found that the SEP intensity is not dominated by locally accelerated ions associated with interplanetary shocks at such energies according to multispacecraft observations.

Furthermore, researchers use numerical simulations and comparison with observations to study the acceleration and transport of SEPs (e.g., [26–34]). To include various transport mechanisms, such as source effects, adiabatic cooling, and perpendicular diffusion, a reservoir of SEPs can be obtained (e.g., [28,35]). In addition, some research obtained the perpendicular and parallel diffusion coefficients with simulations and observations in different longitude and radial distance (e.g., [29,31,36–38]). Moreover, the effects of magnetic cloud (MC) and sheath region on lower-energy SEPs during a ground level enhancement (GLE) event are studied with the comparison of numerical simulations and observations to infer that the enhanced turbulence level and varied magnetic field in sheath-MC can depress the proton intensity of SEP events [39]. In addition, impulsive SEP events often exhibit sudden drops in particle counts known as SEP dropouts, which affect all energy ranges simultaneously [40,41]. Recent observations have revealed the presence of magnetic signatures associated with SEP dropouts, implying that they originate from the local interaction between SEPs and magnetic structures in the solar wind [42,43].

At present, empirical SEP forecasting methods are based on the statistical relationships between the characteristics of associated phenomena and the properties of SEP events. These empirical SEP forecasting methods can be used to predict the main characteristic parameters of an SEP event, such as onset time, peak flux, event-integrated fluence, duration, and the SEP event occurrence. The proton prediction model of NOAA's Space Weather Prediction Center (SWPC) utilizes the flare features to predict the SEP event occurrence [44,45]. The UMASEP scheme based on the lag correlation between strong positive derivatives of X-ray flux and proton flux to forecast the occurrence and the intensity of the first hour of SEP events [46,47]. In addition, state-of-the-art forecasting schemes also include the forecasting solar particle events and flares (FORSPEF) system [48,49], the empirical model for solar proton events real-time alert (ESPERTA) [50,51], and the proton prediction system (PPS) [52,53], etc.

Many of previous studies on SEP events show the statistical relationship between SEP event properties and flare features, which can be further used to predict SEP event properties. SEPs would experience many transport effects, such as adiabatic cooling, magnetic focusing, and perpendicular and pitch-angle diffusion in the expanding and turbulent solar wind. Therefore, these empirical relationships might fail if the interplanetary environments become complex in some events. Furthermore, SEPs in impulsive events are suggested to be generated by some explosive process, e.g., the magnetic reconnection, near the flare site, so that the Reid–Axford profile [54] is widely used to represent the source function of particle injection near the Sun (e.g., [26,35,55]). However, there are few works that study the relationship between the flare features and the source parameters of particle-injection function due to the fact that the particle injection near the flare site is hard to measure.

In this work, with simulations we reproduce the electron intensity-time profiles of five SEP events observed near 1 au by multispacecraft, i.e., ACE, STEREO-A, and STEREO-

B. We obtain the particle-injection source parameters and turbulence level—which are collectively referred to key parameters—of these events by fitting the simulation results to the observations. By analyzing the relationship between the key parameters and event features from observations, we establish the model of key parameters, so that the SEP events can be predicted in simulations by using the model-predicted key parameters. The paper is structured as follows. The observation data and event selection are introduced in Section 2. The simulation model of SEPs is described in Section 3. The simulation results and analyses are presented in Section 4. The model of key parameters and its evaluation are shown in Section 6. The conclusions and discussion are presented in Section 7.

## 2. Data

In this work, we collect near-relativistic electron data from observations of ACE/EPAM [56] and STEREO/SEPT [57] instruments. ACE and twin STEREO spacecraft can provide a wide longitudinal range of observations with nearly the same heliocentric distances to avoid radial gradient effects. The selected energy channels of ACE/EPAM and STEREO/SEPT are 175–315 keV and 195–225 keV, respectively. Here, the effective energy for modeling is set to $E_0 = 210$ keV for both ACE and STEREO (e.g., [33]). We select a list of solar energetic electron events during the period from 2007 to 2014 according to the following criteria: (1) the SEP event observed by multispacecraft should have identification of the source; (2) the corresponding flare information, such as the location and SXR intensity-time profile, is available; (3) the SEP data observed by STEREO do not have strong ion contamination; and (4) the SEP events are not affected noticeably by shocks.

Based on the selection criteria, we have compiled a catalog of 17 solar energetic electron events, of which the characteristics are summarized in Table 1. The events can be classified into two groups, Group I with 5 events and Group II with 12 events, as shown in the first column of Table 1. For Group I, all events are observed at least by two spacecraft simultaneously without large contamination. In contrast, the events in Group II are mostly observed by one of the three spacecraft without large contamination. The date of SEP events is presented in column 2. Columns 3–5 show the onset time, class, and position of flares, which are observed by GOES in the 0.1–0.8 nm wavelength band (http://www.ngdc.noaa.gov/stp/GOES/, accessed on 1 May 2022). Column 6 gives the time-integrated fluence, $F_{SXR}$. Columns 7–8 list the longitudes of STEREO-A and STEREO-B where positive/negative indicates that the spacecraft is in the west/east relative to ACE. Columns 9, 10, and 11 present the solar wind speeds, $V^{sw}$, observed by ACE, STEREO-A, and STEREO-B, respectively, at the time of the flare onset. The instruments to measure solar wind on ACE and STEREO are SWEPAM [58] and PLASTIC [59], respectively. Columns 12–13 give the maximum interplanetary magnetic field (IMF), $B_{max}$, and the maximum fluctuation, $\delta B_{max}$, of the IMF within the day before the flare onset, provided by ACE/MAG [60] with the time resolution of 16 seconds. The ratio of $\delta B_{max}$ to $B_{max}$ is indicated as the magnetic ratio, $r_{max} \equiv \delta B_{max}/B_{max}$, shown in Column 14.

**Table 1.** Collected Group I and Group II SEP events with observed properties.

| Group | Date | Flare Onset | Flare Class | Flare Site | $F_{SXR}$ | $\phi_{STA}$ | $\phi_{STB}$ | $V^{sw}_{ACE}$ | $V^{sw}_{STA}$ | $V^{sw}_{STB}$ | $B_{max}$ | $\delta B_{max}$ | $r_{max}$ |
|---|---|---|---|---|---|---|---|---|---|---|---|---|---|
| | | (UT) | | | (J m$^{-2}$) | (deg) | (deg) | (km s$^{-1}$) | (km s$^{-1}$) | (km s$^{-1}$) | (nT) | (nT) | |
| | 7 February 2010 | 02:20 | M6.4 | N21E10 | $3.8 \times 10^{-2}$ | 65 | −71 | 351 | 509 | 464 | 9.6 | 4.6 | 0.48 |
| | 14 August 2010 | 09:38 | C4.4 | N17W52 | $9.9 \times 10^{-3}$ | 80 | −72 | 442 | 362 | 328 | 5.8 | 1.6 | 0.28 |
| I | 24 February 2011 | 07:23 | M3.5 | N14E87 | $2.0 \times 10^{-2}$ | 87 | −95 | 355 | 318 | 656 | 4.7 | 2.6 | 0.55 |
| | 15 April 2012 | 02:16 | C1.7 | N15E88 | $2.9 \times 10^{-3}$ | 112 | −119 | 505 | 356 | 544 | 4.5 | 2.2 | 0.49 |
| | 22 October 2013 | 21:15 | M4.2 | N04W01 | $7.5 \times 10^{-3}$ | 148 | −142 | 345 | 288 | 297 | 9.5 | 3.8 | 0.40 |

**Table 1.** *Cont.*

| Group | Date | Flare Onset | Flare Class | Flare Site | $F_{SXR}$ | $\phi_{STA}$ | $\phi_{STB}$ | $V^{sw}_{ACE}$ | $V^{sw}_{STA}$ | $V^{sw}_{STB}$ | $B_{max}$ | $\delta B_{max}$ | $r_{max}$ |
|-------|------|-------------|-------------|------------|-----------|--------------|--------------|----------------|----------------|----------------|-----------|------------------|-----------|
|       |      | (UT)        |             |            | (J m$^{-2}$) | (deg) | (deg) | (km s$^{-1}$) | (km s$^{-1}$) | (km s$^{-1}$) | (nT) | (nT) | |
|       | 6 February 2010 | 06:59 | C4.0 | N21E22 | $1.4\times10^{-3}$ | 65 | −71 | 344 | 406 | 483 | 6.2 | 2.2 | 0.35 |
|       | 7 August 2010 | 17:55 | M1.0 | N11E34 | $1.8\times10^{-2}$ | 79 | −72 | 397 | 508 | 337 | 4.6 | 2.2 | 0.48 |
|       | 28 January 2011 | 00:44 | M1.3 | N15W88 | $9.5\times10^{-3}$ | 86 | −93 | 306 | 513 | 681 | 5.7 | 2.8 | 0.49 |
|       | 13 February 2011 | 17:28 | M6.6 | S20E04 | $4.0\times10^{-2}$ | 87 | −94 | 337 | 501 | 323 | 4.4 | 1.7 | 0.39 |
|       | 2 August 2011 | 05:19 | M1.4 | N14W15 | $3.9\times10^{-2}$ | 100 | −93 | 497 | 413 | 674 | 5.0 | 2.5 | 0.50 |
| II | 8 August 2011 | 18:00 | M3.5 | N16W61 | $2.2\times10^{-2}$ | 101 | −93 | 589 | 675 | 280 | 6.1 | 1.8 | 0.30 |
|    | 11 November 2011 | 06:11 | C4.2 | S18W42 | $1.6\times10^{-2}$ | 106 | −103 | 401 | 317 | 436 | 8.4 | 4.3 | 0.52 |
|    | 24 April 2012 | 07:38 | C3.7 | N12E83 | $3.6\times10^{-3}$ | 113 | −118 | 412 | 504 | 337 | 15.8 | 4.1 | 0.26 |
|    | 3 June 2012 | 17:48 | M3.3 | N16E38 | $7.0\times10^{-3}$ | 116 | −117 | 357 | 484 | 346 | 15.7 | 5.9 | 0.38 |
|    | 30 August 2013 | 02:04 | C8.3 | N13E43 | $4.4\times10^{-2}$ | 145 | −138 | 344 | 408 | 335 | 6.1 | 1.8 | 0.30 |
|    | 28 July 2014 | 13:56 | C2.4 | S08E51 | $4.0\times10^{-3}$ | 164 | −162 | 410 | 426 | 289 | 13.0 | 5.9 | 0.45 |
|    | 30 July 2014 | 16:00 | C9.0 | S10E38 | $1.3\times10^{-2}$ | 164 | −162 | 330 | 430 | 324 | 5.7 | 2.6 | 0.46 |

## 3. SEP Transport Model

### 3.1. Focused Transport Equation

The three-dimensional focused transport equation that governs the particle gyrophase-averaged distribution function $f(\boldsymbol{r}, \mu, p, t)$ can be written as [26,35,37,61]

$$
\begin{aligned}
\frac{\partial f}{\partial t} = {} & \nabla \cdot (\boldsymbol{\kappa}_\perp \cdot \nabla f) + \frac{\partial}{\partial \mu}\left(D_{\mu\mu}\frac{\partial f}{\partial \mu}\right) - \left(v\mu\,\hat{\boldsymbol{b}} + \boldsymbol{V}^{sw}\right)\cdot\nabla f \\
& + p\left[\frac{1-\mu^2}{2}\left(\nabla\cdot\boldsymbol{V}^{sw} - \hat{\boldsymbol{b}}\,\hat{\boldsymbol{b}} : \nabla\boldsymbol{V}^{sw}\right) + \mu^2\,\hat{\boldsymbol{b}}\,\hat{\boldsymbol{b}} : \nabla\boldsymbol{V}^{sw}\right]\frac{\partial f}{\partial p} \\
& - \frac{1-\mu^2}{2}\left[-\frac{v}{L} + \mu\left(\nabla\cdot\boldsymbol{V}^{sw} - 3\,\hat{\boldsymbol{b}}\,\hat{\boldsymbol{b}} : \nabla\boldsymbol{V}^{sw}\right)\right]\frac{\partial f}{\partial \mu},
\end{aligned}
\tag{1}
$$

where $\boldsymbol{r}$ is the position in the heliosphere relative to the center of the Sun, $\mu$ is the pitch-angle cosine, $p$ and $v$ are the momentum and speed of particles, respectively, in the solar wind frame, $t$ is the time, the tensor $\boldsymbol{\kappa}_\perp$ represents the perpendicular diffusion coefficient, $D_{\mu\mu}$ is the pitch-angle diffusion coefficient, $\hat{\boldsymbol{b}}$ is a unit vector of the mean magnetic field, $\boldsymbol{V}^{sw} = V^{sw}\,\hat{\boldsymbol{r}}$ is the solar wind velocity in the radial direction, and $L$ is the magnetic focusing length given by $L = \left(\hat{\boldsymbol{b}}\cdot\nabla\ln B_0\right)^{-1}$ with $B_0$ being the magnitude of the background interplanetary magnetic field (IMF). Note that the Parker field is chosen as the IMF with $B_0 = 5$ nT at 1 au and a constant solar wind speed measured by the spacecraft at the time of flare onset. Equation (1) includes many important processes, such as particle streaming along magnetic field lines, adiabatic cooling, magnetic focusing, and perpendicular and pitch-angle diffusion.

### 3.2. Diffusion Coefficients

The pitch-angle diffusion coefficient as a function of $\mu$, which is the pitch-angle cosine of energetic particles, could be written as [62,63]

$$
D_{\mu\mu}(\mu) = \left(\frac{\delta B_{slab}}{B_0}\right)^2 \frac{\pi(s-1)}{4s}\frac{v}{l_{slab}}\left(\frac{R_L}{l_{slab}}\right)^{s-2}\left\{|\mu|^{s-1} + h\right\}\left(1-\mu^2\right),
\tag{2}
$$

where $\delta B_{slab}/B_0$ is the magnetic turbulence level of slab component, $s = 5/3$ is the Kolmogorov spectral index of magnetic turbulence in the inertial range, $l_{slab}$ is the correlation length of $\delta B_{slab}$ and is set to be 0.031 au in this work, $R_L = pc/(|q|B_0)$ is the particle Larmor radius, and the constant $h = 0.01$ is introduced to simulate the particles' ability to scatter through $\mu = 0$ [64,65].

If the pitch-angle distribution of particles is always nearly isotropic, the parallel mean free path $\lambda_\parallel$ can be derived from the pitch-angle diffusion coefficient $D_{\mu\mu}$ [66,67]

$$\lambda_{\parallel} = \frac{3v}{8} \int_{-1}^{+1} \frac{(1-\mu^2)^2}{D_{\mu\mu}} d\mu, \tag{3}$$

and the parallel diffusion coefficient $\kappa_{\parallel}$ can be expressed as $\kappa_{\parallel} = v\lambda_{\parallel}/3$.

The perpendicular diffusion coefficient is taken from the nonlinear guiding center theory with the approximation in the analytic forms [68,69]

$$\kappa_{\perp} = \frac{1}{3}v \left[ \left( \frac{\delta B_{2D}}{B_0} \right)^2 \sqrt{3}\pi \frac{s-1}{2s} \frac{\Gamma\left(\frac{s}{2}+1\right)}{\Gamma\left(\frac{s}{2}+\frac{1}{2}\right)} l_{2D} \right]^{2/3} \times \lambda_{\parallel}^{1/3} \left( \mathbf{I} - \hat{\boldsymbol{b}} \hat{\boldsymbol{b}} \right), \tag{4}$$

where $\delta B_{2D}/B_0$ is the magnetic turbulence level of 2D component, $l_{2D}$ is the correlation length, $\Gamma$ is the gamma function, and $\mathbf{I}$ is a unit tensor. The ratio of slab turbulence energy to 2D turbulence energy is suggested to be 20:80 [70], so that

$$\frac{\delta B_{\text{slab}}}{B_0} = \frac{\sqrt{5}}{5} \frac{\delta B}{B_0}, \tag{5}$$

$$\frac{\delta B_{2D}}{B_0} = \frac{2\sqrt{5}}{5} \frac{\delta B}{B_0}, \tag{6}$$

where $\delta B/B_0$ is the turbulence level of the IMF. According to Weygand et al. [71,72], the correlation scale ratio of slab to 2D can be set to 2.6, so that $l_{2D} = 0.012$ au.

### 3.3. Source Function of Particle Injection

The source of SEP events is assumed to be located around the flare site, $(\theta_0, \phi_0)$, and the energetic particles are modeled to be injected at the inner boundary of the simulations (0.05 au). The Reid–Axford profile [54] is widely used to represent the source distribution function of particle injection for SEP events (e.g., [26,34,35,55,73])

$$f_s(r \le 0.05 \text{ au}, \theta, \phi, E_k, t) = \frac{C}{t} \cdot \frac{E_k^{-\gamma}}{p^2} \exp\left( -\frac{\tau_c}{t} - \frac{t}{\tau_L} \right) \xi(\theta, \phi), \tag{7}$$

where $r$ is the heliocentric distance, $C$ is a normalization constant, $\gamma = 3$ is the spectral index of the injected particles, $E_k$ is the kinetic energy of the source particle, $\tau_c$ and $\tau_L$ are the rise and decay timescales of the particle-injection profile, respectively, and $\xi(\theta, \phi)$ represents the source region,

$$\xi(\theta, \varphi) = \begin{cases} a(\theta, \varphi) & |\theta - \theta_0| \le \Delta\theta \text{ and } |\phi - \phi_0| \le \Delta\varphi, \\ 0 & \text{otherwise}, \end{cases} \tag{8}$$

where $\theta_s$ and $\phi_s$ are the half-widths of the particle source in the latitudinal and longitudinal directions, respectively, and $\theta$ and $\phi$ are the longitude and latitude of the position of concern, respectively. In addition, $\phi_s$ and $\theta_s$ are set to have the same value in this work. It is noted that the SEP source can be over a wide range [34] because of some mechanisms, e.g., the random walk of magnetic field lines [74].

The peak differential flux of the particle injection can be derived from Equation (7)

$$j_s^{\text{max}}(E_k) \equiv p^2 f_s^{\text{max}} = \frac{C}{t_{\text{max}}} \frac{E_k^{-\gamma}}{p^2} \exp\left( -\frac{\tau_c}{t_{\text{max}}} - \frac{t_{\text{max}}}{\tau_L} \right) \tag{9}$$

with

$$t_{\text{max}} = \frac{\sqrt{\tau_L^2 + 4\tau_c\tau_L} - \tau_L}{2}, \tag{10}$$

where $t^{\text{max}}$ is the peak time of the particle injection. We can define a characteristic peak intensity of the particle source as follows,

$$j_m \equiv j_s^{\text{max}}|_{E_k = E_0} = \frac{C}{t_{\text{max}}} \frac{E_0^{-\gamma}}{p_0^2} \exp\left( -\frac{\tau_c}{t_{\text{max}}} - \frac{t_{\text{max}}}{\tau_L} \right), \tag{11}$$

where $p_0$ is the corresponding momentum of the effective energy, $E_0$, of the spacecraft measurements. In this work, we choose the characteristic peak intensity, $j_m$, of the particle source as a basic parameter of the source function, so $C$ can be expressed as

$$C = j_m t_{\max} \frac{p_0^2}{E_0^{-\gamma}} \exp\left(\frac{\tau_c}{t_{\max}} + \frac{t_{\max}}{\tau_L}\right). \tag{12}$$

Therefore, the particle-injection source function, Equation (7), has four source parameters, namely, the rise timescale $\tau_c$, the decay timescale $\tau_L$, the half width $\theta_s$, and the characteristic peak intensity $j_m$.

We use the time-backward Markov stochastic process theory to solve Equation (1) with the boundary condition given by Equation (7) to obtain the SEP distribution function $f$ at the spacecraft, so that the source parameters and the magnetic turbulence level can be obtained by fitting the simulated differential flux $j = p_0^2 f$ to the observed one. Note that the outer boundary is set to 50 au. The detailed description of the application of the method to study the transport of SEPs can be found in the literature [26,35,75].

## 4. Best-Fit Simulation Results for SEP Events in Group I

In the following, we reproduce the multispacecraft observed electron intensity-time profiles of the SEP events in Group I by numerically solving the focused transport equation. We try different parameters of the source and magnetic turbulence level to obtain the best-fit parameters listed in Table 2, $\tau_c$, $\tau_L$, $\theta_s$, $j_m$, and $\frac{\delta B}{B_0}$, so that the simulations fit the observations the best.

**Table 2.** Best-fit parameters for the Group I SEP events.

| Date | $\tau_c$ (min) | $\tau_L$ (min) | $\theta_s$ (deg) | $j_m$ $((cm^2\ sr\ s\ MeV)^{-1})$ | $\frac{\delta B}{B_0}$ |
|---|---|---|---|---|---|
| 7 February 2010 | 66 | 66 | 40 | $7.59 \times 10^4$ | 0.272 |
| 14 August 2010 | 101 | 101 | 48 | $5.87 \times 10^4$ | 0.210 |
| 24 February 2011 | 230 | 274 | 45 | $4.87 \times 10^4$ | 0.457 |
| 15 April 2012 | 130 | 158 | 87 | $7.68 \times 10^3$ | 0.353 |
| 22 October 2013 | 115 | 115 | 65 | $1.89 \times 10^4$ | 0.353 |

Figure 1a–e show the locations of the three spacecraft, which are all at the heliocentric radial distance close to 1 au, and the associated flares, for the five Group I SEP events seen from the north ecliptic pole. The purple arrows indicate the longitudes of the flares as seen from the Earth. The spiral curves represent the Parker magnetic field lines connecting each of the three spacecraft to the Sun, calculated by using a constant solar wind speed measured locally by each spacecraft at the onset of the flare. For any position with solar distance $r$, latitude $\theta$, and longitude $\phi$, the magnetic footpoint with latitude $\theta_f$ and longitude $\phi_f$ can be determined with

$$
\begin{aligned}
\theta_f &= \theta, \\
\phi_f &= \phi + \frac{\Omega r}{V^{sw}},
\end{aligned}
\tag{13}
$$

where $\Omega = 2\pi/T$ is the angular speed of solar rotation with the solar rotation period $T = 25.4$ day.

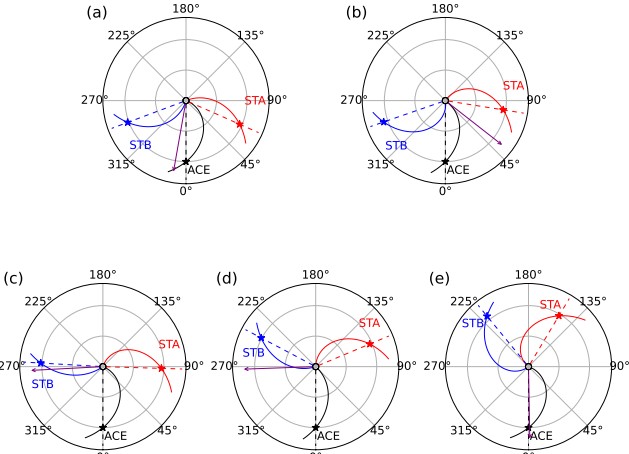

**Figure 1.** Schematic representation of the source regions and spacecraft locations for the five Group I SEP events (**a**) 7 February 2010, (**b**) 14 August 2010, (**c**) 24 February 2011, (**d**) 15 April 2012, and (**e**) 22 October 2013 as seen from the north ecliptic pole. The black, red, and blue spirals are the nominal IMF lines for ACE, STEREO-A (STA), and STEREO-B (STB) spacecraft connected to the Sun, respectively, and the corresponding dashed lines show the projected longitudes of these spacecraft. The purple arrow indicates the flare longitude.

The observed electron intensity-time profiles and the results from simulations with the best-fit parameters are presented as the solid and dashed lines, respectively, in the top panels of Figure 2a–e for the five SEP events, and the bottom panels of Figure 2a–e show the corresponding election injection profiles given by Equation (7). The purple vertical lines indicates the time of the flare onset.

Figure 2a presents the results of the event on 7 February 2010. Although all the three spacecraft observations show particle enhancement after the flare eruption, the particle source of STEREO-A may be from other flares according to the explanation given by the STEREO SEPT solar electron event list (http://www2.physik.uni-kiel.de/stereo/downloads/sept_electron_events.pdf, accessed on 20 March 2023). Indeed, it is hard to reproduce the electron intensity-time profiles of the three spacecraft well simultaneously, so that we only fit the simulation results to the observations from ACE and STEREO-B. Figure 2b shows the results for the event on 14 August 2010. We can see that the simulation results successfully reproduce most features of the observations for this case. Figure 2c presents the results of the event on 24 February 2011. Figure 2d shows the results for the event on 15 April 2012. In the cases of 24 February 2011 and 15 April 2012, the simulation results for STEREO-A and STEREO-B agree well with observations, and there were no enhancements for both simulations and observations of ACE. Figure 2e shows the results for the event on 23 October 2013. In this case, the rise phase of the STEREO-B observation and the decay phase of the ACE observation are not fitted well, which may be due to some coronal or interplanetary effects. In addition, there were no enhancements for both simulations and observations of STEREO-A. All in all, among the five SEP events, only the event on 14 August 2010 was clearly observed by all the three spacecraft, ACE, STEREO-A, and STEREO-B simultaneously, and the observations can be reproduced by simulations with the source location determined from the solar flare. Generally speaking, the simulation model can generally reproduce the multispacecraft observations with spacecraft in different locations as shown in Figure 1.

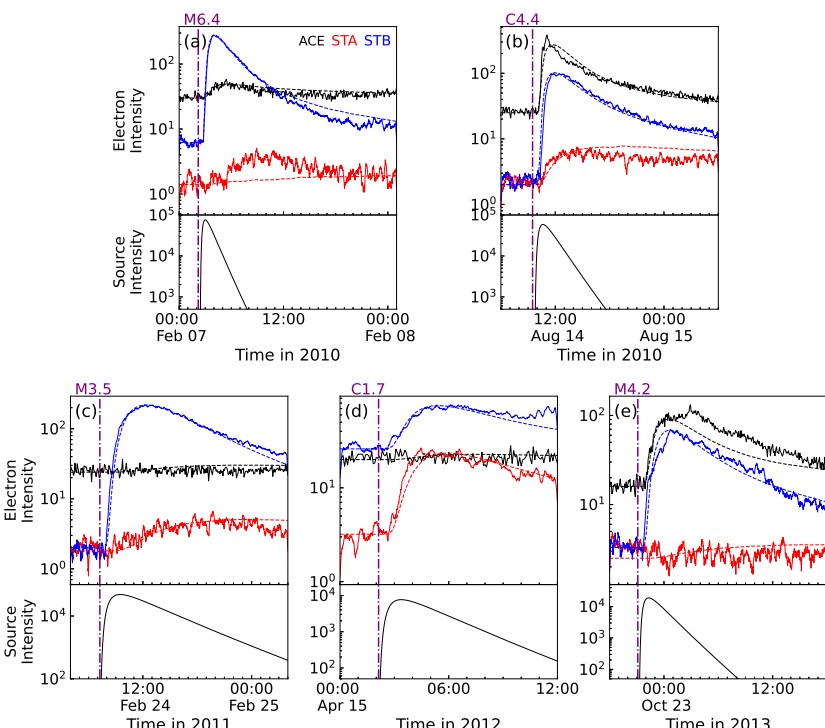

**Figure 2.** The upper parts of the five panels show the comparison between the simulated electron intensity-time profiles (dashed lines) with the observed ones (solid lines) for the five Group I SEP events, while the lower parts present the corresponding intensity-time profiles of particle injection. The black, red, and blue colors denote the results for ACE, STEREO-A, and STEREO-B, respectively. The vertical dot-dashed line indicates the flare onset.

## 5. Model for Key Parameters

In order to use our numerical model to calculate the flux of SEP for the prediction purpose, we need the parameters of the energetic particle source, i.e., $\tau_c$, $\tau_L$, $\theta_s$, and $j_m$, and the turbulence level $\frac{\delta B}{B_0}$, which are collectively referred to the key parameters. The ratio of the maximum IMF fluctuation $\delta B_{max}$ to the maximum IMF $B_{max}$ observed within one day before the flare onset is indicated as the magnetic ratio, $r_{max} \equiv \delta B_{max}/B_{max}$. Next, we construct the empirical model of the key parameters as the function of the observed properties of flare, i.e., $F_{SXR}$, and the magnetic ratio $r_{max}$.

### 5.1. Correlations among the Key Parameters

It is possible that there is a correlation between two key parameters, so we check it with the regression method (e.g., [10,76,77]). In Figure 3a, the best-fit rise timescale of particle source, $\tau_c$, is shown as a function of the best-fit turbulence level, $\frac{\delta B}{B_0}$, for simulations. In this figure, the red dashed line shows the linear regression between $\tau_c$ and $\frac{\delta B}{B_0}$ with CC = 0.85 and the level of statistical significance, $p = 6.8\%$. We can see that there is a strong correlation between them. From the linear regression we have

$$\tau_c = k_c \left( \frac{\delta B}{B_0} \right) + \tau_{c0}, \tag{14}$$

where $k_c = 5.6 \times 10^2$ min and $\tau_{c0} = -55.72$ min. The regression is reasonable since the increasing of the turbulence level indicates a stronger event, which may lead to the increasing of the rise timescale of particle source. We can use this model to get $\tau_c$ for the purpose of prediction.

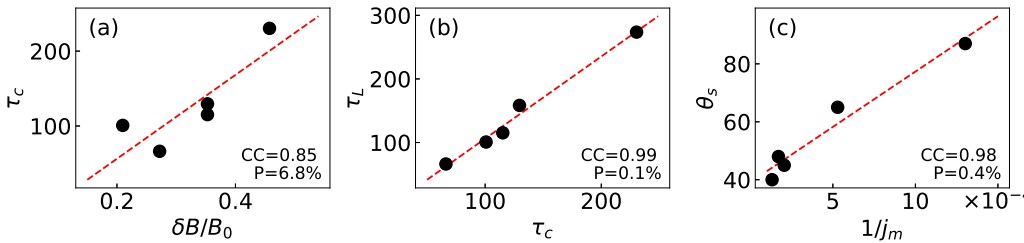

**Figure 3.** Linear regressions between (**a**) the rise timescale, $\tau_c$, and the turbulence level, $\delta B/B_0$, (**b**) the decay timescale, $\tau_L$, and the rise timescale, $\tau_c$, and (**c**) the half width, $\theta_s$, and the reciprocal of the characteristic peak intensity, $j_m$.

In addition, we find that the best-fit decay timescale, $\tau_L$, of the particle source has strong correlations with $\tau_c$, as shown in Figure 3b, with CC = 0.99 and $p = 0.1\%$

$$\tau_L = k_L \tau_c + \tau_{L0}, \tag{15}$$

where $k_c = 1.29$ min and $\tau_{L0} = -23.09$ min.

Moreover in Figure 3c, the red dashed line shows the linear regressions between the half-width, $\theta_s$, and the reciprocal of the characteristic peak intensity, $1/j_m$, with $CC = 0.98$ and $p = 0.4\%$. Therefore, there is a strong correlation between $\theta_s$ and $1/j_m$. From the linear regression, we have

$$\theta_s = \theta_{s0}\left(\frac{j_0}{j_m} + 1\right), \tag{16}$$

where $\theta_{s0} = 39.1°$ and $j_0 = 9.71 \times 10^3$ $(\text{cm}^2 \text{ sr s MeV})^{-1}$. A possible interpretation of the anticorrelation between $\theta_s$ and $j_m$ is that the extension of the particle source would lead to the reduction of the strength of the source if the total number of seed particles remains basically unchanged. For the prediction purpose, we can use this model to get $\theta_s$ if we have a model for $j_m$.

Therefore, we need to obtain the models for the turbulence level, $\frac{\delta B}{B_0}$, and the characteristic peak intensity, $j_m$ of the particle source. To do so, we find the statistical relationship between these parameters listed in Table 2 and the observed event features in Table 1 for the Group I SEP events.

*5.2. Models for the Best-Fit Key Parameters*

In Figure 4a, the best-fit characteristic source peak intensity, $j_m$, is shown as a function of the observed flare fluence, $F_{SXR}$. In this figure, the red dashed line shows the linear regression between $j_m$ and $F_{SXR}$, with CC = 0.84 and $p = 7.8\%$. It is suggested that there is a strong correlation between $j_m$ and $F_{SXR}$. Therefore, from the linear regression we have

$$j_m = k_j F_{SXR} + j_{m0}, \tag{17}$$

where $k_j = 1.69 \times 10^6$ $(\text{sr s MeV}^2)^{-1}$ and $j_{m0} = 1.56 \times 10^4$ $(\text{cm}^2 \text{ sr s MeV})^{-1}$. The correlation is reasonable because higher intensity of flare indicates stronger particle source. We can use this model to get $j_m$ for the prediction purpose. It is noted that the significance levels, $p$, for all the cases in Figure 4 (including the cases shown below) is greater than the usual 5% level; however, it is still under the 10% level that is sometimes used.

Similarly, we study the relationship between one of the best-fit key parameters for simulations, the turbulence level, $\frac{\delta B}{B_0}$, and the magnetic ratio, $r_{max}$, from observations. In Figure 4b, $\frac{\delta B}{B_0}$ is shown as a function of $r_{max}$. In the figure, the red dashed line shows the linear regression between $\frac{\delta B}{B_0}$ and $r_{max}$, with CC = 0.81 and $p = 9.5\%$. From the linear regression we have

$$\frac{\delta B}{B_0} = k_b r_{max} + \delta_0, \tag{18}$$

where $k_b = 0.71$ and $\delta_0 = 0.015$. We can use this model to get $\frac{\delta B}{B_0}$ for the purpose of prediction.

All other key parameters, $\tau_c$, $\tau_L$, and $\theta_s$, can be obtained from Equations (14)–(16).

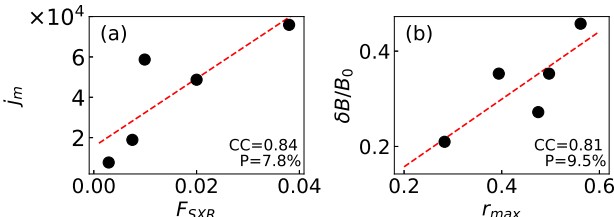

**Figure 4.** Linear regressions between (**a**) the best fit of characteristic peak intensity, $j_m$, and the observed flare fluence, $F_{SXR}$, and (**b**) the best-fit turbulence level, $\delta B / B_0$, and the ratio of the maximum IMF fluctuation to the maximum IMF, $r_{max}$, observed within the day before the flare onset.

## 6. Simulation Results for SEP Events in Group II without Fitting

The parameters of particle source, $\tau_c$, $\tau_L$, $\theta_s$, $j_m$, and turbulence level, $\delta B / B_0$, can be obtained from the models described above. We simulate the 12 SEP events in Group II with the numerical code using these models. Since these simulation results are obtained without fitting to the observations, we indicate these results as predictions. We compare the predicted SEP intensities with the observations near 1 au to test the prediction method.

Similar to Figure 1, Figure 5 shows the locations of footpoints of the three spacecraft and the flare longitude for the 12 Group II events. It is shown that the distributions of spacecraft locations are also diverse, so that the 12 events can evaluate the prediction model comprehensively. The flare properties and magnetic field parameters of the 12 events are listed in Table 1. The modeling results of the source parameters and turbulence levels provided by Equations (14)–(18) are listed in Table 3 for the 12 Group II events. Figure 6 presents the comparison of prediction results and observations for the 12 events. The upper part of each panel in Figure 6 shows the comparison between the predicted electron intensity-time profiles (dashed lines) and the observed ones (solid lines) near 1 au, while the lower part of each panel presents the predicted source intensity-time profiles. It is shown that the predicted intensity-time profiles are not in good agreement with all the observations.

To further illustrate the potential applicability of the prediction model, we compare the predicted peak flux and event-integrated fluence with the observed ones near 1 au because the two parameters can represent the intensity of particle radiation. The comparisons of peak flux and event-integrated fluence between predictions and observations are presented in Figure 7a,b, respectively. The black, red, and blue solid circles represent the results for ACE, STEREO-A, and STEREO-B, respectively, and the numbers labeled around the circles denote the sequence number of the 12 Group II SEP events. The maximum intensity reached shortly after the onset of the event is chosen as the peak intensity since some observations are contaminated, thus showing nontypical intensity-time profiles. Note that if there is no particle enhancement in the observation or simulation, the corresponding background flux is chosen as the peak flux and used to calculate the event-integrated fluence. The event-integrated fluence is the time integration of the flux. For the observed fluence, the integration starts at the onset of the event and ends at the time when the flux declines to the background level or encounters a new particle enhancement in the decay phase. Note that the end times of the fluence observed by the three spacecraft are chosen to be the same. For comparison, the integration time interval of the predicted fluence is set to the same as that of the observed fluence. The purple solid line in each panel is the linear regression between the observation and prediction in double logarithmic coordinates with the CC reported in the figure. It is shown that both the peak flux and fluence have a good correlation between the observations and predictions. The gray dashed line in each panel indicates where the prediction equals the observation. It is shown that the purple solid lines are close to the gray dashed lines, indicating that the predictions can well reproduce the observations. To

quantify the goodness of the prediction results, the root mean squared error (RMSE) is also reported in the figure and given by

$$\text{RMSE} = \sqrt{\frac{\sum_{i=1}^{N}(x_i - y_i)^2}{N}},\qquad(19)$$

where $x_i$ and $y_i$ represent the predictions and observations, respectively, and $N$ is the total number of observations or simulations. Therefore, the prediction errors of peak flux and fluence are approximately 0.4 orders of magnitude.

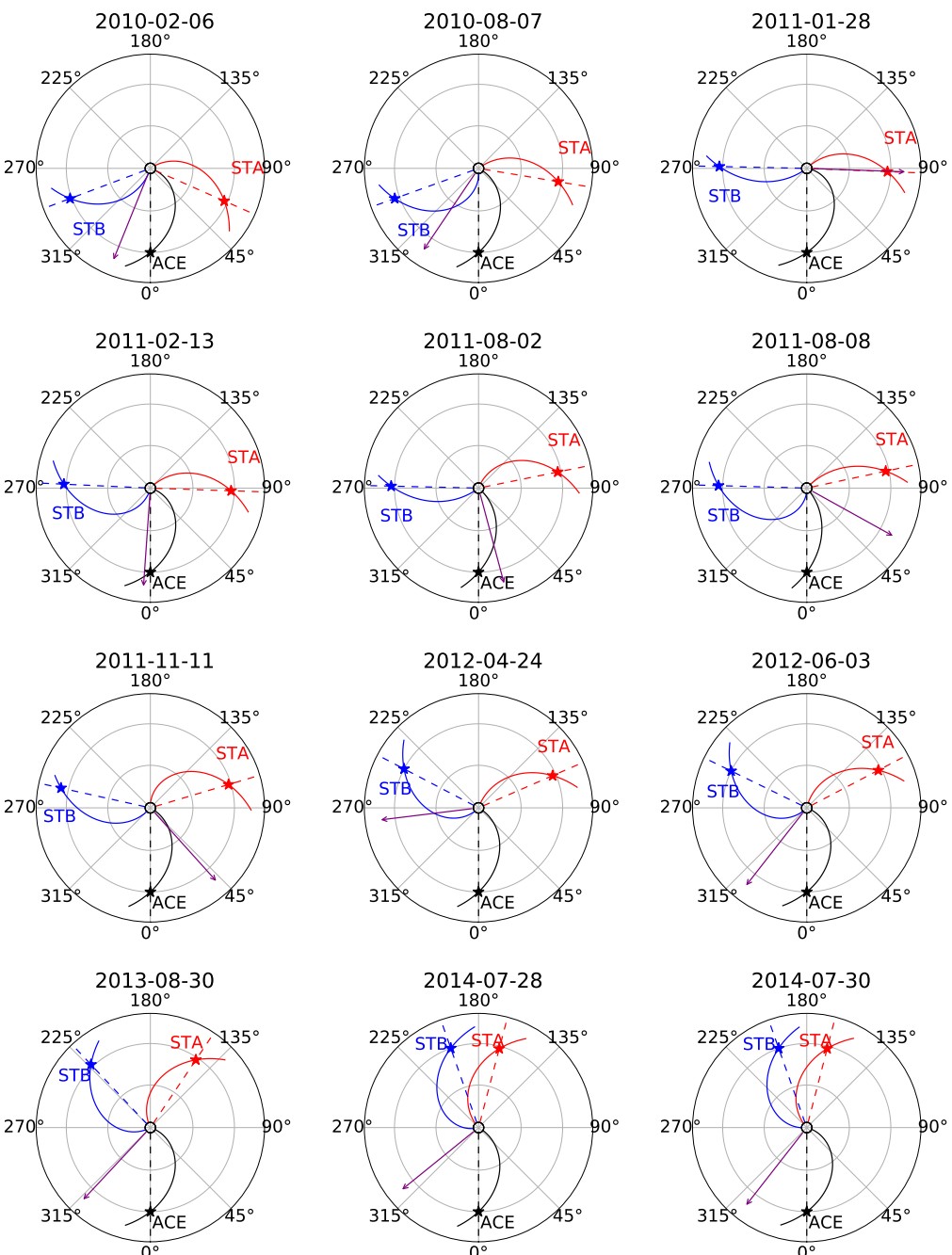

**Figure 5.** Same as Figure 1 but for the 12 Group II SEP events.

**Table 3.** Modeling results of source parameters and turbulence levels.

| Date | $\tau_c$ (min) | $\tau_L$ (min) | $\theta_s$ (deg) | $j_m$ ((cm² sr s MeV)⁻¹) | $\frac{\delta B}{B_0}$ |
|---|---|---|---|---|---|
| 6 February 2010 | 95 | 100 | 60 | $1.80\times10^4$ | 0.270 |
| 7 August 2010 | 147 | 167 | 47 | $4.60\times10^4$ | 0.362 |
| 28 January 2011 | 146 | 165 | 51 | $3.17\times10^4$ | 0.360 |
| 13 February 2011 | 106 | 113 | 44 | $8.32\times10^4$ | 0.288 |
| 2 August 2011 | 154 | 176 | 44 | $8.15\times10^4$ | 0.374 |
| 8 August 2011 | 72 | 69 | 46 | $5.28\times10^4$ | 0.227 |
| 11 November 2011 | 158 | 180 | 48 | $4.26\times10^4$ | 0.381 |
| 24 April 2012 | 57 | 51 | 57 | $2.17\times10^4$ | 0.202 |
| 3 June 2012 | 101 | 107 | 53 | $2.74\times10^4$ | 0.280 |
| 30 August 2013 | 71 | 69 | 43 | $9.00\times10^4$ | 0.226 |
| 28 July 2014 | 134 | 150 | 56 | $2.24\times10^4$ | 0.339 |
| 30 July 2014 | 134 | 150 | 49 | $3.76\times10^4$ | 0.340 |

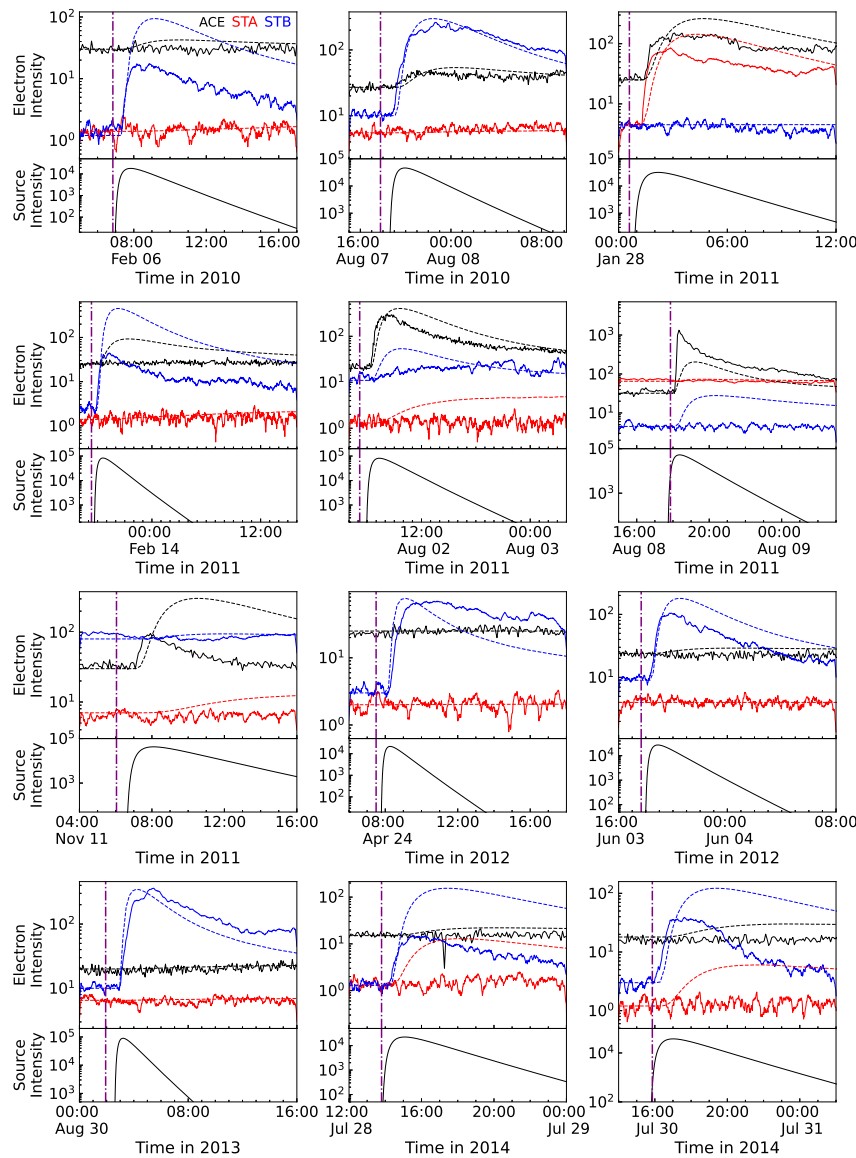

**Figure 6.** Same as Figure 2 except that the source intensities of these 12 Group II SEP events are predicted by the key parameters model.

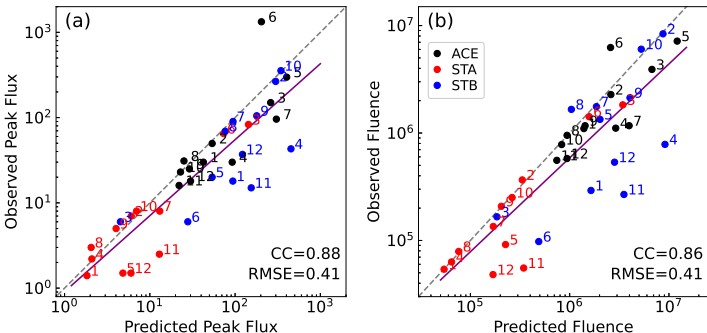

**Figure 7.** Comparison of predicted and observed (**a**) peak flux and (**b**) fluence for 12 Group II SEP events. The numbers labeled around the circles are the sequence number of the 12 Group II SEP events. The purple lines are the linear regressions in double logarithmic coordinates, while the gray dashed lines indicate where the prediction equals to the observation.

## 7. Conclusions and Discussion

In this paper, we compile a catalog of 17 solar energetic electron events observed by at least one of the three spacecraft, namely, ACE, STEREO-A, and STEREO-B, during the period from 2007 to 2014. The selected energy channels of ACE/EPAM and STEREO/SEPT are 175–315 keV and 195–225 keV, respectively, and their effective energies are set to 210 keV. These SEP events can be classified into two groups. The events in Group I are observed by at least two spacecraft simultaneously, while the events in Group II are observed by one spacecraft only or observed by multispacecraft, but only a small portion of the measurements are without large contamination. We reproduce the observed intensity-time profiles of the five Group I SEP events in simulations because multispacecraft observations can provide more restrictions. The simulations are carried out by solving the three-dimensional focused transport equation, Equation (1), with the boundary condition given by the source function, Equation (7), of particle injection, so that for the five events we can obtain the magnetic turbulence level, $\frac{\delta B}{B_0}$, and the source parameters, i.e., rise timescale $\tau_c$, decay timescale $\tau_L$, the half-width $\theta_s$, and characteristic peak intensity $j_m$, which are referred to the key parameters.

Then, we statistically analyze the relationship among the key parameters themselves and between the key parameters and the observed properties, i.e., the time-integrated fluence of flare, $F_{SXR}$, and the magnetic ratio, $r_{max}$, observed by ACE within one day before the flare onset. The rise time, $\tau_c$ is found to be correlated with the turbulence level, $\frac{\delta B}{B_0}$, the decay timescale $\tau_L$ is correlated with $\tau_c$, and the half-width, $\theta_s$ is anticorrelated with the characteristic peak intensity, $j_m$. In addition, it is found that the characteristic peak intensity, $j_m$, is positively correlated with the flare fluence, $F_{SXR}$. The turbulence level, $\delta B/B_0$, is found to be correlated with the magnetic ratio, $r_{max}$, observed by ACE within one day before the flare onset. Therefore, the model of the key parameters is established.

To evaluate the key parameter models, we simulate the electron intensity-time profiles of the 12 Group II events by using the input parameters predicted by the source parameters model. Though the predicted electron intensity-time profiles cannot fit the observed ones well, the peak flux and event-integrated fluence, which are important parameters for estimating the radiation doses, can be predicted well. The key parameters can be obtained at the onset of SEP events, so that the SEP transport simulation with the key parameters model can be used to predict the peak flux and event-integrated fluence of SEP events, and further the radiation hazards of SEPs can be estimated.

Our study reveals that the magnetic field turbulence level ($\frac{\delta B}{B_0}$) exhibits a narrow range of variation. We attempted to use a constant value (e.g., an average value) of $\frac{\delta B}{B_0}$ in our numerical simulations, but we cannot obtain good results. The reason might be that the slight change of $\frac{\delta B}{B_0}$ can make the results change a great deal. This narrow range of $\frac{\delta B}{B_0}$ on the solar wind turbulence may be caused from the fact that the events we select were not

significantly affected by IP shocks. However, if the events are significantly affected by the IP shock, the range of variation in $\frac{\delta B}{B_0}$ may be wide.

The solar maximum of solar cycle 25 is suggested to occur in October 2024 (95% confidence interval is from February 2023 to September 2026) [78], so that the Parker Solar Probe (PSP) [79,80] and Solar Orbiter [81,82] would observe many SEP events in the next few years. The combination of PSP, Solar Orbiter, and near-Earth spacecraft can provide comprehensive observations of SEP events since these spacecraft can spread in different radial distances, longitudes, and latitudes. With more SEP events included in Group I and Group II, the key parameters model can be further validated and improved.

The electron intensity-time profiles of Group II events cannot be predicted well, which is partly due to the fact that the interplanetary environments of Group II events are more complex than that of Group I events. Further studies can simulate the SEP events by including more realistic interplanetary or coronal magnetic fields (e.g., [39,83–86]).

**Author Contributions:** Data curation, L.L.; investigation, L.L., G.Q. and S.C.; methodology, L.L., G.Q., S.W. and Y.W.; writing—original draft preparation, L.L. and S.W.; writing—review and editing, G.Q., L.L., Y.W. and S.C. All authors have read and agreed to the published version of the manuscript.

**Funding:** This research was funded by research grant of Natural Science Foundation of China Grant Numbers: 42074206 and U1831129.

**Institutional Review Board Statement:** Not applicable.

**Informed Consent Statement:** Not applicable.

**Data Availability Statement:** The data for this paper are available at the Space Physics Data Facility (SPDF) (https://spdf.gsfc.nasa.gov/, accessed on 20 March 2023), and the CDAW CME catalog (http://cdaw.gsfc.nasa.gov/CME_list/, accessed on 20 March 2023).

**Acknowledgments:** We thank the ACE/EPAM, STEREO/SEPT, and GOES/XRS teams for providing the data used in this paper. The ACE data are provided by the ACE Science Center, the STEREO data are provided by the STEREO Science Centers, and the GOES data are provided by the NOAA. Figures were prepared with Matplotlib [87].

**Conflicts of Interest:** The authors declare no conflict of interest.

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
