# Peer review of "Modeling of the Magnetic Turbulence Level and Source Function of Particle Injection from Multiple SEP Events"

_magnetochemistry, doi:10.3390/magnetochemistry9040091_

Round 1

Reviewer 1 Report

This article examines the fluence and time profiles of energetic electrons in large SEP events. The authors identified 5 parameters as the key parameters and using a formalism developed by some of the co-authors, fitted multiple events by varying these key parameters. 

The method is sound and the explanation of the procedure is clear.  I support its publication after minor changes.

Line 194 mentions that the best fitting results for the 5 events in group I is obtained by varying the key parameters.  What is the metrics for determining the fitting to be the best? How much freedom can one change one parameter and then compensated by varying another (other) parameter(s)?

Figure 4 is quite interesting. How does it look if the 12 group-II events are also included?  In principle, for the dB/B parameter, one can use some kind of average value prior to the event, instead of relying on this regression result. Is that right?  The fact that dB/B only vary slightly in Table III seems to suggest that it is an insensitive parameter, although dB and B can vary significantly in these events.  What is the implication of this narrow range of dB/B on the solar wind turbulence?

The reference list contains 94 items. For a research article rather than a review articles, some may not be necessary.  This is only a minor suggestion.

Reviewer 2 Report

Comments on the paper “Modeling of the Magnetic Turbulence Level and Source

Function of Particle Injection from Multiple SEP Events” by Lele Lian, Gang Qin, Shuangshuang Wu, Yang Wang and Shuwang Cui

This paper presents a new model to predict the Solar Energetic Particles fluxes detected near 1 AU based on five fitted key parameters for particle injection source parameters and turbulence level in the solar wind. The paper is developed from observations from ACE, STEREO A and STEREO B spacecraft, for 5 impulsive SEP events when the soft X-ray flare source region was identified on the sun. The model is then tested based on other 12 events. Even if the number of events is limited, the results are interesting, and the paper is very well written. I suggest the publication once the minor comments below are addressed.

Minor comments:

Line 60: It would be worth to mention here that impulsive SEP events show often abrupt modulations of particles counts called SEP dropouts, affecting all energies at the same time (Mazur et al. 2000, Gosling et al. 2004). More recent observations highlighted the presence of magnetic signatures associated with SEP dropouts, suggesting that they are generated locally by the interaction of SEP with magnetic structures in the solar wind (Trenchi et al., 2013a ; Trenchi et al., 2013b).

Line 85: It seems that a verb is missing. It should be or: "We reproduce" or "we are reproducing”.

Caption of Figure 2: “The upper parts of the 5 panels show the comparison between the simulated electron intensity-time profiles (solid lines) with the observed ones (dashed lines)”. I believe the description of simulated Vs observed time profiles is the opposite. I.e., simulated intensity are the dashed lines, the observed the solid lines. Please verify.

Figures 2 & 6: I know it is reported in the legend of figure 2, and legend of figure 6 makes reference to Figure 2. But, just for simplicity, the Authors may consider to add a small legend with colours – s/c: red = STA ; blue = STB; black = ACE.

Line 300 / Figures 5 & 6: Comparing Figures 5 and 6, it seems that, for event events, the predictions of your model better represent the observations if considering the spacecraft that has the foot-point of the Parker field line on the sun closer to the flare (e.g. on 2010-02-06 and 2010-08-07  good agreement for STB; on 2011-01-28 good agreement on ACE and STA; and so on). On the contrary, the predictions are worst when the s/c foot-point on the sun is farther away from the flare site, when the model predicts higher fluxes than observed (e.g. on 2011-08-08 for STB ; 2014-07-30 for STA).  I suggest adding a brief discussion to mention this feature (if the Authors believe that it could be relevant), and try to explain why.

References:

-        Mazur, J. E., Mason, G. M., Dwyer, J. R., et al. 2000, ApJL, 532, L79

-        Gosling, J. T., Skoug, R. M., McComas, D. J., & Mazur, J. E. 2004, ApJ, 614, 412

-        Trenchi, L., Bruno, R., Telloni, et al: Solar Energetic Particle Modulations Associated with Coherent Magnetic Structures, Astrophys. J., 770, doi:10.1088/0004-637X/770/1/11, 2013.

-        Trenchi, L., Bruno, R., D’Amicis, R. et al.: Observations of IMF coherent structures and their relationship to SEP dropout events, Ann. Geophys., 31, 1333–1341, 2013
